# Mindful Eating, Nutrition Knowledge, and Weight Status among Medical Students: Implications for Health and Counseling Practices

**DOI:** 10.3390/nu16121894

**Published:** 2024-06-16

**Authors:** Denis Mihai Serban, Sorin Ursoniu, Radu Dumitru Moleriu, Ancuta Mioara Banu, Costela Lacrimioara Serban

**Affiliations:** 1Department of Obstetrics-Gynecology, Discipline of Obstetrics-Gynecology II, “Victor Babes” University of Medicine and Pharmacy, 2 Eftimie Murgu Square, 300041 Timisoara, Romania; denis.serban@umft.ro; 2Department of Functional Sciences, Discipline of Public Health, Center for Translational Research and Systems Medicine, “Victor Babes” University of Medicine and Pharmacy, 2 Eftimie Murgu Square, 300041 Timisoara, Romania; sursoniu@umft.ro (S.U.); costela.serban@umft.ro (C.L.S.); 3Department of Functional Sciences, Discipline of Medical Informatics and Biostatistics, “Victor Babes” University of Medicine and Pharmacy, 2 Eftimie Murgu Square, 300041 Timisoara, Romania; 4Department 2, Discipline of Maxillo-Facial Surgery, Faculty of Dental Medicine, “Victor Babes” University of Medicine and Pharmacy, 2 Eftimie Murgu Square, 300041 Timisoara, Romania; banu.ancuta@umft.ro

**Keywords:** mindfulness of eating, nutrition knowledge, BMI, weight, obesity, medical students

## Abstract

Academic stress and transitioning to young adulthood can lead medical students to develop inadequate eating habits, affecting both their physical and mental well-being and potentially compromising their ability to offer effective preventive counseling to future patients. The primary objective of this study is to analyze the levels of mindful eating and nutrition knowledge in Romanian medical students and their associations with various sociodemographic variables. Additionally, we explore the relationship between mindful eating and nutrition knowledge while accounting for sociodemographic factors and examine the impact of these factors on excess weight. Significant predictors of excess weight include high weight gain in the past year (OR = 15.8), the mindful eating questionnaire (MEQ) score (OR = 0.131), male gender (OR = 2.5), and being in the clinical years of medical school (OR = 2.2). While nutrition knowledge levels do not directly impact weight status, they share a 4% common variance with mindfulness in multivariate analysis. Notably, high weight gain is independent of the levels of mindful eating, but univariate testing links it to disinhibition and emotional response, components of mindfulness. Mindful eating stands out as independently associated with both nutrition knowledge and excess weight among medical students. Thus, interventions to address obesity should consider incorporating mindfulness training to enhance food intake awareness and improve weight management outcomes in Romanian medical students.

## 1. Introduction

The rigors and challenges of medical school pose a serious threat to the well-being of aspiring doctors. Research indicates that over 50% of medical students experience severe distress, which can lead to a decline in quality of life, suicidal ideation, and an increase in dropout rates [1]. Academic stress, heavy workloads, the need for continuous learning, and exposure to illness and death during their education put medical students at high risk for developing mental health issues, including eating disorders [2]. The risk of eating disorders is additionally further increased by the transitional nature through young adulthood [3].

The relationship between academic stress and unhealthy eating behavior varies among individuals. While some may resort to unhealthy eating habits as a coping mechanism, others may adopt alternative strategies such as restrained eating [4,5,6]. It is important to recognize that individuals often make food choices unconsciously, leading to unintentional overeating and weight gain [7]. Mindful eating, defined as a non-judgmental awareness of physical and emotional sensations while eating, which promotes a more attentive and purposeful approach to food intake, has been shown to be beneficial in addressing various health issues, including eating disorders and stress [8]. A comprehensive understanding of nutrition plays a pivotal role in ensuring optimal health by facilitating informed food choices [9,10,11].

For medical students, the failure to maintain healthy eating practices could lead to detrimental effects on their physical and mental health while studying medicine. Medical students must prioritize healthy eating habits for their well-being and the benefit of their future patients [12,13,14]. Research suggests that medical students who maintain healthier personal practices are likelier to provide effective preventive counseling to their patients as physicians [15,16].

The primary objective of this study is to analyze the levels of mindful eating and nutrition knowledge in medical students from Romania and their associations with various sociodemographic variables. Secondary objectives include exploring associations between mindful eating and nutritional knowledge while controlling for other sociodemographic variables and examining the impact of mindful eating and general nutritional knowledge on excess weight while controlling for sociodemographic effects.

## 2. Materials and Methods

### 2.1. Study Population

Based on data from a previous systematic review [17], which indicated a response rate of 69% among medical students with a standard deviation of ±20.8%, we extended invitations to 140 students per academic year (years one to six). This approach aimed to secure a sample size exceeding 500 participants from the total enrollment of 2800 students in the Romanian section of the Faculty of Medicine at Victor Babes University of Medicine and Pharmacy in Timisoara, Romania. Utilizing an online calculator, we estimated the margin of error in sampling for 500 participants at a 95% confidence level to be 5% [18].

### 2.2. Procedure

Firstly, this investigation received approval from the Scientific Research Vice-rectorate of Victor Babes University of Medicine and Pharmacy Timisoara, Romania, under number 23660 from 18 October 2022.

A total of 140 students per academic year were invited to participate in this project. Upon approaching students, the study’s aims and expected results were communicated. To incentivize participation, it was promised that students would receive a general nutrition knowledge score upon form submission.

For data collection, we used an online instrument based on Google Forms. The instrument included an informed consent statement on its initial page. Participants were required to acknowledge their consent before proceeding to the next page, and the completion of all questions was necessary to submit the form, ensuring no incomplete responses. Consequently, the final database was completely free of any missing data.

The response rate varied across academic years, with a maximum of 80.7% observed in the first year and a minimum of 51.4% in the sixth year of medical studies. The mean response rate was 62.1%, with a standard deviation of 10.6%.

The median competition time of the survey was 16 min with an interquartile range of 2.4 min, a minimum of 12.8, and a maximum of 21.2 min.

### 2.3. Instruments

The investigation included two previously validated instruments: Framson’s mindful eating questionnaire [19], validated in Romanian by Serban et al. [20], and Kliemann’s general nutrition knowledge questionnaire [21] validated in Romanian by Putnoky et al. [22], along with sociodemographic questions.

### 2.4. Measurements and Data Management

For the mindful eating questionnaire, the score was calculated as previously described by Framson [19] for each of the 28 items using a 4-point Likert-type scale, with the following response variants “Never/Rarely”, “Sometimes”, “Often”, and “Usually/Always”. Answers to items 3, 4, 5, 8, 10, 12, 14, 15, 16, 20, 21, 22, 23, 24, 25, and 26 received scoring from 1 to 4. For items 1, 2, 6, 7, 9, 11, 13, 17, 18, 19, 27, and 28, the scoring was reversed. The maximum possible score is 4. The 28-item questionnaire has five domains: awareness, distraction, disinhibition, emotional response, and external cues. Each domain was calculated as the mean of the item score, and the summary score was the mean of the five domains. Higher scores are associated with a more mindful eating approach and lower scores with a less mindful approach. The questionnaire is available as Supplementary Material with the initial validation [20]. This instrument was used in previous research among student populations by Moore et al. [23].

The general nutrition knowledge questionnaire, developed by Kliemann [21], was used to investigate nutrition knowledge. The instrument comprises 88 questions each with 1 correct answer, totaling 88 possible points. Each question included an “I do not know” option. The questionnaire has 4 sections: Section 1 Expert recommendations (maximum 18 points), Section 2 Food groups (maximum 36 points), Section 3 Healthy food choices (maximum 13 points), and Section 4 Diet, disease, and weight associations (max 21 points). For the total score, each correct answer received one point. For wrong answers or if the responder chose the “I do not know” option, no points were either subtracted or added. For this research, the Romanian version of the questionnaire was used [22]. The questionnaire is available as Supplementary material with the initial validation [22]. This instrument was previously used with different research questions in student populations [24,25].

The survey utilized a variety of questions to gather information from participants, including their gender, year of medical studies, age, weight, height, and perceived level of stress. The stress level was rated on a scale of 1 to 10, with higher scores indicating greater levels of stress. Students were also asked to rate their health status using a five-point scale ranging from excellent to bad. Weight trends over the past year were evaluated using a 5-point scale that ranged from losing more than 5 kg, losing between 1 and 5 kg, or having an unchanged weight (−1 kg to 1 kg) to gaining between 1 and 5 kg and gaining more than 5 kg.

Using this data, the body mass index (BMI) was calculated by dividing weight (in kg) by height (in meters) squared. Certain sociodemographic variables were recoded, and the BMI was categorized as weight status: underweight, normal weight, overweight, or obese. It was further divided into those with excess weight (BMI ≥ 25.0 kg/m^2^) and those without (BMI < 25.0 kg/m^2^). Health status was categorized as excellent or very good perceived health versus all other values. The perceived levels of stress were spread across tertiles and then dichotomized into the highest tertile of stress versus lower levels. Weight trends during the last year were categorized as gaining more than 5 kg versus all other options. Clinical years were determined by dichotomizing the year of study into preclinical (years 1–3) and clinical (years 4–6).

Categorical data are presented as percentages and counts. Continuous data were tested for normality with the Kolmogorov–Smirnoff test and if normally distributed were presented as means and standard deviations (SDs). For non-normally distributed variables, for central tendency and spread, the median and the interquartile range (IQR) were used. The *t*-test and Mann–Whitney test were used to compare continuous variables by dichotomous factors when normality and respective non-normality were assumed. For the prediction of eating mindfulness and the presence of excess weight, linear and respective logistic regressions were used. The mean general nutrition knowledge scores obtained by students (overall and by preclinical/clinical years) were compared to the mean populational values obtained from a Romanian general populational sample obtained during the validation study [22]. The *p* values for all hypothesis tests were two-sided, and statistical significance was set at *p* < 0.05, except for *p*-values calculated in Tables 2 and 3 which were adjusted for the false discovery rate (FDR) with the Benjamini–Hochberg procedure using an online tool (https://tools.carbocation.com/FDR) (accessed on 12 January 2024). This adjustment was necessary due to multiple comparisons using sections and overall scores of each of the mindful eating questionnaire and general nutrition knowledge questionnaire. All analyses were conducted using IBM SPSS 21.

## 3. Results

Females represented 75.8% (395) of the group, and 61% (318) of the group attended clinical years (years 3–6 of medical school). The median age for the whole group was 22.4 years with an interquartile range of 1.9, a minimum of 18 years, and a maximum of 30 years. The median BMI was 21.6 kg/m^2^ with an interquartile range of 3.5 kg/m^2^, a minimum of 15.7 kg/m^2^, and a maximum of 42.1 kg/m^2^. A percentage of 40.5% (211) perceive themselves to have excellent or very good health status, and 36.7% (191) were in the highest tertile of stress. Excess weight, namely BMI levels of 25 kg/m^2^ or more, was reported by 18.5% (96) of the participants, and 4.8% (25) of the entire group gained more than 5 kg in the last year (Table 1).

### 3.1. Mindfulness of Eating in Students

Our results indicate that of a maximum of four achievable points, indicating the highest level of mindfulness, our sample obtained a mean score for mindfulness of eating of 2.8 ± 0.3. Using univariate analysis, higher levels of mindfulness were registered by participants reporting excellent or very good health status compared with those reporting low levels of health status, low levels of stress, compared with those reporting high levels of stress, no excess weight compared with participants with excess weight, and for those without weight gain over 5 kg during the last year compared participants without this trait (Table 2). Women and people reporting excellent or very good health status had better scores in awareness (Table 2). Men, people reporting excellent or very good health status, and people with lower levels of stress had better scores in distraction (Table 2). Women, people reporting low levels of stress, those with no excess weight, and those without weight gain over 5 kg during the last year had better scores in disinhibition (Table 2). Women, people reporting lower health status and people with higher levels of stress, those with excess weight, and those with over 5 kg weight gain during last year had higher levels of emotional response (Table 2). Men had higher levels of external cues (Table 2).

### 3.2. General Nutrition Knowledge Questionnaire

In Table 3 are presented total and section knowledge scores for the general nutrition knowledge questionnaire in a sample of 521 responders by different demographic characteristics of the sample. The highest possible value of the knowledge score is 88 points. The knowledge score had a median (IQR) of 59.0 (12.5). Students in clinical years (years 3–6 of medical school) had higher scores per total and sections (Table 3). The other comparisons were not statistically significant after the FDR adjustment (Table 3).

When compared to a Romanian general population sample [22] which had a median (IQR) of 58.0 (17), the whole sample was not statistically significant, *p* = 0.245, but stratifying the analysis in preclinical and clinical students obtained significantly lower (*p* < 0.001) and respective significantly higher (*p* < 0.001) levels of nutrition knowledge as compared to the general population.

### 3.3. Prediction Model for the Mindfulness of Eating

Table 4 contains significant demographic predictors of the mindful eating score, calculated using linear regression. Since higher levels of the score were related to higher levels of the mindfulness of eating, these were significantly related to higher levels of nutrition knowledge (Beta = 0.240), excess weight (Beta = −0.181), female gender (Beta = 0.136), high levels of perceived health status (Beta = 0.108), and lower levels of stress (Beta = −0.078), when controlling for the effects of high weight gain in the last year and being in the clinical years of medical school (3rd to 6th year of medical school).

### 3.4. Prediction Model for Excess Weight

Table 5 includes the significant contributors and coefficients for the prediction of excess weight in the study sample by using logistic regression. Significant predictors were high weight gain during the last year (OR = 15.8 95% CI = 5.9–42.5), MEQ score (OR = 0.131 95% CI =0.05–0.36), male gender (OR = 2.5; 95%CI 1.4–4.2), and clinical years in medical school (OR = 2.2 95% CI = 1.2–4.0). The levels of stress, perception of health status, and the level of nutrition knowledge were not significant predictors but are accounted for in the model.

## 4. Discussion

In this study, we explored the relationship between the mindfulness of eating, nutrition knowledge, and excess weight among Romanian medical students, an area of inquiry not previously investigated in this population to the best of our knowledge.

Our findings suggest that while there is no significant relation between nutrition knowledge and excess weight among medical students, mindful eating appears to be associated with both nutrition knowledge and excess weight. Controlling for various variables in a multivariate model, nutrition knowledge accounted for 4% of the unique variance in mindful eating (obtained as part of the linear regression). Specifically, a 1 SD increase in the nutrition knowledge score (equivalent to 12.5 points) corresponded to a 0.240 SD increase in the mindful eating score (or 0.072 points). However, the clinical significance of this association may be limited due to the study’s design. Conversely, mindless eating, which is the opposite of mindful eating, emerged as a predictor of excess weight in the multivariate model, with each 1-point decrease in the mindful eating score associated with a 7.6 times increase in the likelihood of having excess weight (1/OR = 1/0.131 = 7.6).

Previous research has separately examined the associations between the mindfulness of eating, nutrition knowledge, and excess weight. However, the simultaneous exploration of these factors within the same context is novel. Studies investigating the association between nutrition knowledge and obesity have yielded mixed results, potentially influenced by sampling and sociodemographic factors. While some studies reported nonsignificant associations [22,26], others found significant links [27,28]. Longitudinal studies have indicated that higher nutrition knowledge may lead to reduced BMI levels [29,30,31], yet the mindful aspects of eating have not been extensively addressed in these studies.

Similar to our findings, previous research has established a negative and significant relationship between BMI and mindful eating. Cross-sectional studies conducted with medical students [23] and the general population [20,32] have consistently shown that higher BMI correlates with lower levels of mindful eating. Intervention studies have further demonstrated that increasing mindfulness can lead to weight loss [33]. However, there is a scarcity of the literature examining the connection between nutrition knowledge and mindful eating. Kurtipek et al. [34] conducted a cross-sectional study investigating this association and found that undergraduate sports students exhibited a shared variance level of less than 1%. Our study involving medical students identified a higher level of shared variance (4%).

Previous research has also linked mindfulness to reduced risk-taking behaviors [35,36,37], while stress and low quality of life have been associated with mindless eating [20,38,39]. Our data on medical students corroborate these associations, highlighting the role of stress and quality of life in influencing eating behaviors. Increasing mindfulness has been demonstrated to mitigate stress and enhance overall quality of life and well-being [40,41,42].

Our study conducted among medical students in Romania revealed a mean mindfulness level of 2.8 +/− 0.3. This finding aligns with prior research conducted by Moore et al. [23], which reported a mean mindfulness level of 2.89 ± 0.32 among medical students. Additionally, in a study by Choi, nurses exhibited a mean mindfulness level of 2.75 ± 0.28 [43].

Our findings reveal that medical students demonstrate higher nutrition knowledge during clinical years compared to their preclinical counterparts and the general population. However, there remains room for improvement, as preclinical students exhibit lower knowledge levels than the general population. Comparing the knowledge scores of medical students with those of the general population helps to highlight specific knowledge gaps and educational needs within the student group. Surprisingly, despite these disparities, Medical Universities in Romania do not offer specific nutrition courses for medical students. Aggarwal et al. [44] found that medical doctors generally have lower levels of nutrition knowledge, particularly regarding expert recommendations.

Our research indicates that students in clinical years are 2.2 times more likely to report excessive weight compared to their nonclinical peers, a trend consistent with the findings of Leventer-Roberts et al. [45], where third-year medical residents were 2.2 times more likely to be overweight than first-year residents. The demanding nature of medical training poses challenges in maintaining healthy habits such as balanced nutrition, adequate sleep, and regular physical activity, all of which are associated with weight gain [44] and burnout [46].

A weight gain exceeding 5 kg within the last year poses significant health risks. Students reporting such weight gain are 15.8 times more likely to have excess weight (Table 5). In our multivariate model, high weight gain remains independent of mindful eating (Table 4). However, univariate testing reveals a significant association with disinhibition and emotional response (Table 2). Disinhibited eating, characterized by a lack of control for overeating, particularly in response to palatable foods or negative emotions, has been linked to impulsivity and maladaptive eating behaviors, as suggested by recent research utilizing functional magnetic resonance imaging in healthy adolescents [47].

Emotional eating, on the other hand, refers to eating triggered by negative emotions. Reviews and meta-analyses examining the effects of mindfulness interventions on individuals with excess weight acknowledge the roles of disinhibition and emotional eating in weight gain over time [48,49].

Research indicates that integrating mindfulness principles into interventions can yield positive outcomes, such as promoting mindful eating behaviors and impacting various weight-related factors. Participants in these interventions may experience enhancements in emotional eating, binge eating, and satiety cues while fostering a healthier relationship with food and body image [8]. Moreover, mindfulness-based interventions can contribute to weight control by addressing multifaceted mechanisms involving changes in eating behavior, stress reduction, and enhanced self-regulation, aligning with a holistic approach to weight management. Clearly, mindfulness-based interventions hold promise in supporting healthier habits and fostering a positive relationship with food [33,50].

However, it is important to acknowledge potential limitations in the generalization of these findings, due to the single-center approach of this research. Additionally, a selection bias may be present, as individuals with an increased interest in nutrition and healthy living might be more inclined to participate. Furthermore, relying on self-reported weight and height poses a limitation to the study’s accuracy. The mean population’s general nutrition knowledge score, used for comparisons with the students’ mean value, was obtained as part of a separate research project [22] conducted by the same research group.

## 5. Conclusions

In our study, mindful eating emerged as an independent factor associated with both nutrition knowledge and excess weight among medical students. Despite clinical year students demonstrating better nutrition knowledge, they exhibited a higher excess weight prevalence than their preclinical counterparts. Moreover, students who reported gaining more than 5 kg in the last year displayed tendencies towards mindless eating, characterized by high levels of disinhibition and emotional eating.

These findings suggest that interventions aimed at reducing the obesity burden should consider incorporating mindfulness training to enhance the awareness of food intake. Such training holds the potential for improving both short-term and long-term weight management outcomes among Romanian medical students.

## Figures and Tables

**Table 1 nutrients-16-01894-t001:** Sociodemographic descriptives of the study participants (*n* = 521).

Sociodemographic Factors		% (Count)
Gender	M	24.2% (126)
	F	75.8% (395)
Excellent or very good perceived health	No	59.5% (310)
	Yes	40.5% (211)
The highest tertile of perceived stress	No (1–6 points) Median (IQR)	63.3% (330) 6 (3)
	Yes (7–10 points) Median (IQR)	36.7% (191) 8 (1)
BMI categories	Underweight	12.3% (64)
	Normal weight	69.3% (361)
	Overweight	15.4% (80)
	Obese	3.1% (16)
Over 5 kg weight gain during last year	No	95.2% (496)
	Yes	4.8% (25)
Clinical years (3rd–6th year of medical school)	No	39.0% (203)
	Yes	61.0% (318)
Year of study in medical school	1st	21.7% (113)
	2nd	17.3% (90)
	3rd	15.5% (81)
	4th	14.4% (75)
	5th	17.3% (90)
	6th	13.8% (72)
Age (years)	Median (IQR)	22.4 (1.9)

**Table 2 nutrients-16-01894-t002:** Mindful eating scores and subscales by sociodemographic factors (*n* = 521).

Sociodemographic Factors		MEQ Total	Awareness	Distraction	Disinhibition	Emotional Response	External Cues
Total score		2.8 ± 0.3	2.8 ±0.5	2.8 ±0.6	3.0 ±0.6	2.6 ±0.5	2.4 ± 0.5
Gender	M	2.7 ± 0.3	2.7 ± 0.6	2.9 ± 0.6	2.8 ± 0.7	2.8 ± 0.5	2.3 ± 0.5
	F	2.8 ± 0.3	2.9 ± 0.5	2.7 ± 0.6	3.1 ± 0.6	2.5 ± 0.5	2.4 ± 0.5
	*p*-value	**0.001**	**0.004**	**0.018**	**<0.001**	**<0.001**	**0.023**
Excellent or very good perceived health	No	2.7 ± 0.2	2.8 ± 0.5	2.7 ± 0.6	3.0 ± 0.6	2.6 ± 0.5	2.4 ± 0.5
	Yes	2.8 ± 0.3	2.9 ± 0.5	2.9 ± 0.6	3.0 ± 0.6	2.7 ± 0.5	2.4 ± 0.5
	*p*-value	**0.004**	**0.022**	**0.001**	0.776	**0.026**	0.545
The highest tertile of perceived stress	No	2.8 ± 0.3	2.8 ± 0.5	2.8 ± 0.6	3.0 ± 0.6	2.7 ± 0.5	2.4 ± 0.5
	Yes	2.7 ± 0.3	2.9 ± 0.5	2.7 ± 0.6	2.9 ± 0.6	2.5 ± 0.6	2.5 ± 0.5
	*p*-value	**0.020**	0.057	**0.003**	**0.017**	**<0.001**	0.074
Excess weight	No	2.8 ± 0.3	2.8 ± 0.5	2.8 ± 0.6	3.1 ± 0.6	2.6 ± 0.5	2.4 ± 0.5
	Yes	2.6 ± 0.3	2.9 ± 0.5	2.7 ± 0.6	2.6 ± 0.6	2.4 ± 0.5	2.5 ± 0.5
	*p*-value	**<0.001**	0.932	0.141	**<0.001**	**<0.001**	0.057
Over 5 kg weight gain during last year	No	2.8 ± 0.3	2.8 ± 0.5	2.8 ± 0.6	3.0 ± 0.6	2.6 ± 0.5	2.4 ± 0.5
	Yes	2.6 ± 0.2	2.8 ± 0.4	2.7 ± 0.7	2.6 ± 0.6	2.4 ± 0.6	2.6 ± 0.7
	*p*-value	**0.009**	0.758	0.382	**<0.001**	**0.020**	0.259
Clinical years	No	2.8 ± 0.3	2.8 ± 0.5	2.7 ± 0.6	3.0 ± 0.6	2.6 ± 0.5	2.4 ± 0.5
	Yes	2.8 ± 0.3	2.9 ± 0.5	2.8 ± 0.6	3.0 ± 0.6	2.6 ± 0.5	2.4 ± 0.5
	*p*-value	0.919	0.357	0.222	0.184	0.337	0.239

Values represent means and standard deviations (SDs). *p*-values were adjusted for multiple comparisons using a false discovery rate tool. Values in bold are considered statistically significant after false discovery rate adjustment.

**Table 3 nutrients-16-01894-t003:** General nutritional knowledge scores and section scores by sociodemographic factors (*n* = 521).

		Total Score of General Nutritional Knowledge Questionnaire	Expert Recommendations	Food Groups	Healthy Food Choices	Diet, Disease and Weight Associations
Overall questionnaire		59.0 (12.5)	11.0 (3.0)	23.0 (6.0)	10.0 (3.0)	16.0 (4.0)
Gender	M	61.0 (16.0)	11.0 (3.0)	23.5 (7.0)	9.0 (3.0)	15.0 (4.0)
	F	58.0 (12.0)	11.0 (3.0)	22.0 (6.0)	10.0 (3.0)	16.0 (4.0)
	*p*-value	0.360	0.961	0.016	0.054	0.889
Excellent or very good perceived health	No	58.0 (12.0)	11.0 (3.0)	22.0 (6.0)	10.0 (3.0)	15.0 (4.0)
	Yes	61.0 (13.0)	11.0 (3.0)	23.0 (6.0)	10.0 (3.0)	16.0 (4.0)
	*p*-value	0.014	0.060	0.046	0.325	0.040
The highest tertile of perceived stress	No	59.0 (12.0)	12.0 (3.0)	23.0 (6.0)	10.0 (3.0)	16.0 (4.0)
	Yes	59.0 (13.0)	11.0 (3.0)	22.0 (5.0)	10.0 (3.0)	15.0 (4.0)
	*p*-value	0.156	0.111	0.161	0.154	0.297
Excess weight	No	59.0 (12.0)	11.0 (3.0)	22.0 (7.0)	10.0 (3.0)	15.0 (4.0)
	Yes	60.0 (13.5)	12.0 (3.0)	23.0 (6.0)	10.0 (3.0)	16.0 (4.0)
	*p*-value	0.272	0.839	0.244	0.816	0.125
Over 5 kg weight gain during last year	No	59.0 (12.5)	11.0 (3.0)	23.0 (6.0)	10.0 (3.0)	16.0 (4.0)
	Yes	60.0 (11.0)	12.0 (3.0)	22.0 (7.0)	10.0 (2.0)	15.0 (3.0)
	*p*-value	0.672	0.348	0.809	0.417	0.975
Clinical years	No	54.0 (12.0)	11.0 (3.0)	21.0 (5.0)	9.0 (3.0)	14.0 (4.0)
	Yes	62.0 (9.0)	12.0 (3.0)	24.0 (5.0)	10.0 (2.0)	17.0 (4.0)
	*p*-value	**<0.001**	**<0.001**	**<0.001**	**<0.001**	**<0.001**

Values represent medians and interquartile range (IQR). *p*-values were calculated using the Mann–Whitney test. *p*-values were adjusted for multiple comparison using a false discovery rate tool. Values in bold were considered statistically significant after false discovery rate adjustment.

**Table 4 nutrients-16-01894-t004:** Sociodemographic predictors of mindful eating scores.

	Unstandardized Coefficients		Standardized Coefficients	95.0% Confidence Interval for B		*p*-Value
	B	Std. Error	Beta	Lower Bound	Upper Bound	
Female gender (yes/no)	0.085	0.026	0.136	0.033	0.136	0.001
Excellent or very good perceived health (yes/no)	0.059	0.023	0.108	0.014	0.103	0.011
Perceived stress (10 steps ordinal variable)	−0.010	0.005	−0.078	−0.021	−0.001	0.04
Excess weight (yes/no)	−0.125	0.031	−0.181	−0.185	−0.064	<0.001
General nutrition knowledge score	0.007	0.001	0.240	0.004	0.009	<0.001

Test applied: linear regression; dependent variable: mindful eating questionnaire (MEQ) score (higher scores meaning higher mindfulness of eating); independent variables: Gender (F vs M.), excellent or very good perceived health (yes = 1 vs. no = 0), perceived stress (higher the score meaning higher levels of stress), excess weight (yes = 1 vs. no = 0), over 5 kg weight gain during last year (yes = 1 vs. no = 0), clinical years (yes = 1 vs. no = 0), general nutrition knowledge score (a higher score meaning higher levels of knowledge).

**Table 5 nutrients-16-01894-t005:** Sociodemographic predictors of excess weight.

Predictors	B	S.E.	OR	95% C.I. for OR		*p*-Value
				Lower	Upper	
Male gender (yes/no)	0.896	0.270	2.449	1.442	4.159	0.001
High weight gain during last year (yes/no)	2.762	0.503	15.835	5.904	42.471	<0.001
In clinical years in medical school (yes/no)	0.780	0.301	2.182	1.209	3.938	0.010
MEQ total score	−2.033	0.515	0.131	0.048	0.359	<0.001

The test applied: logistic regression; dependent variable: the presence of excess weight; independent variables: male gender (yes = 1 vs. no = 0), excellent or very good perceived health (yes = 1 vs. no = 0), perceived stress (a higher score meaning higher levels of stress), over 5 kg weight gain during last year (yes = 1 vs. no = 0), clinical years (yes = 1 vs. no = 0), general nutrition knowledge score (a higher score meaning higher levels of knowledge), mindful eating score (higher scores meaning a higher mindfulness of eating).

## Data Availability

The datasets used and/or analyzed during the current study are available from the corresponding author on reasonable request.

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
