# Peer review of "Mindful Eating, Nutrition Knowledge, and Weight Status among Medical Students: Implications for Health and Counseling Practices"

_nutrients, 2024, doi:10.3390/nu16121894_

Round 1
Reviewer 1 Report
Comments and Suggestions for Authors
Thank you for the opportunity to review this manuscript. The study focused on mindful eating and nutrition knowledge of medical students and how they are associated with several sociodemographic variables.
Introduction:
11. Objectives between abstract and introduction do not match. Because of that, it is unclear what the exact objective of this study is.
22. There is a lack of defined concepts that are being utilized in the study. For example, mindful eating, weight status, and excess weight. Please provide clear definitions of these terms.
Methods:
33. Reference #17 is identified as a meta-analysis, but it is not a meta-analysis. It is a systematic review.
44. Forcing the response for all questions could lead to inaccurate data collection as the participants may simply answer questions quickly to just get through the survey to finish. Please provide context around the general length of the survey. E.g. How long did it take the participants to complete the survey?
55. What scores on the General Nutrition Knowledge Questionnaire are considered to be knowledgeable vs not knowledgeable about nutrition?
66. How was health status calculated/collected? Why only two categories for health status?
77. Why were the scores of the Knowledge questionnaire compared to population sample? Please provide some context and how this fits into the objectives of the study.
88. What is the highest score that can be achieved for the mindfulness of eating questionnaire??
99. What statistical test was used when adjusting for false discovery rate? The website used for the test was provided, but the exact statistical test must be identified.
Results
110. Line 164 –How do the results of 2.8+/- 0.3 compare to what can be achieved?
111. Line 174 last sentence – does not align with data in Table 2.
112. Table 2 used a “false discovery rate tool” to adjust for multiple comparisons. What is the tool that was used? This should be states in the methods.
113. Line 180-182 – sentence needs clarification/proof reading.
114. Lines 190-194 – Why was this analysis done and how does it contribute to the objective of the study?
Table 4:
115. State the type of data analysis that was performed in this table.
116. Why are both unstandardized and standardized coefficients reported? Only standardized coefficients are discussed in the results, so it is unclear why unstandardized coefficients are included. Please include p values for the regression.
117. Please explain the rationale for using high weight gain in the last year and being in clinical years were as control variables.
118. When refereeing to the MEQ, the foot note states “higher scores meaning lower mindfulness of eating"- while in the methods it is states that “higher scores are associated with more mindful eating approach”. This discrepancy should be rectified.
Table 5
119. Was logistic regression performed for the data in this table? If so, can the data be represented as adjusted OR and 95%CI, and a p value?
Discussion
220. A new concept of mindless eating is brought up without providing a definition of it. At what point (score) in the MEQ is the mindless eating applicable?
221. Line 231 – Where does the 4% come from?
222. Line 237 – How was the value of "7.6 times" calculated?
Comments on the Quality of English LanguageUse of English language is fine throughout the manuscript.
Author Response
Please find the answer to the comments raised by yourself during review in attachment.

Reviewer 2 Report
Comments and Suggestions for Authors
This is a cross-sectional study to examine how mindful eating and nutrition knowledge may influence body weight of medical students. However, mindful eating and nutrition knowledge are two different aspects, i.e., being mindful does not mean being knowledgeable in the healthfulness of own diet, vice versa. It is unclear how both aspects are related, and there has been no comparison between medical students and the general population. Besides, no dietary data is collected, so it’s uncertain whether being more mindful or more knowledgeable in eating lead to a healthier diet, hence a better body weight. Sample size is relatively small (~500) and representativeness is affected. The aforementioned issues would affect the validity of findings.
Comments on the Quality of English LanguageNil
Author Response
Please find the answers the the issues raised during review in the attachment
